# 2D topological matter from a boundary Green's functions perspective: Faddeev-LeVerrier algorithm implementation

**Miguel Alvarado⋆ and Alfredo Levy Yeyati†**

Departamento de Física Teórica de la Materia Condensada C-V,
Condensed Matter Physics Center (IFIMAC) and Instituto
Nicolás Cabrera, Universidad Autónoma de Madrid,
E-28049 Madrid, Spain

⋆ miguel.alvarado@uam.es , † a.l.yeyati@uam.es

## Abstract

Since the breakthrough of twistronics a plethora of topological phenomena in correlated systems has appeared. These devices can be typically analyzed in terms of lattice models using Green's function techniques. In this work we introduce a general method to obtain the boundary Green's function of such models taking advantage of the numerical Faddeev-LeVerrier algorithm to circumvent some analytical constraints of previous works. We illustrate our formalism analyzing the edge features of a Chern insulator, the Kitaev square lattice model for a topological superconductor and the Checkerboard lattice hosting topological flat bands. The efficiency and accuracy of the method is demonstrated by comparison to recursive Green's function calculations.

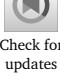

# 1 Introduction

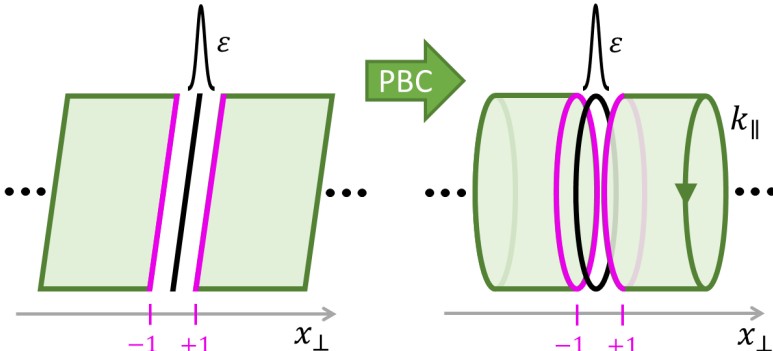

Figure 1: Cylindrical geometry obtained by applying periodic boundary conditions (PBC) along the direction parallel to the boundary in a 2D plane, where $x_\perp$ denotes a coordinate in the perpendicular direction measured in units of the lattice constant. In this geometry there is a well defined momentum $k_\parallel$ and the open boundaries at $x_\perp = \pm 1$ (magenta lines) are obtained by adding a localized impurity line with an amplitude $\varepsilon \to \infty$ (black line) at $x_\perp = 0$. The impurity line breaks the translational symmetry in the $x_\perp$−direction and opens two boundaries in the bulk infinite system.

In recent years, due to the appearance of twistronics [1, 2] and specially since the discovery of the special properties of twisted bilayer graphene at the magic angle [3, 4], there is a renewed interest in 2D topological materials exhibiting different phases of matter (e.g. superconductivity, magnetism, nematicity, etc). In these systems new phenomena arise from the combination of strong interactions and topology.

These circumstances claim for a flexible unified theoretical framework going beyond idealized minimal models to account for interactions, strongly correlated behaviour, spatial inhomogeneities or hybrid devices. Several techniques have been developed to describe these new 2D platforms with special emphasis in the topological properties of open boundaries, like exact Hamiltonian diagonalization of finite systems, wave matching in finite scattering regions [5], some analytical techniques to derive effective boundary Hamiltonians [6] or the complementary approaches provided by $T$-matrix and Green's function formalisms [7–9]. Nevertheless, methods based in exact diagonalization of microscopic Hamiltonians may require huge computational capabilities with information on several model parameters and generally, they provide only numerical results with, in some cases, little or no insight in the underlying physics. For these reasons theoretical mesoscopic descriptions of intermediate complexity which could give us access not only to discrete surface modes but also to a well-defined continuum of excitations are of great interest.

In this work we focus in the calculation of the boundary Green's function (bGF) for such systems, which encodes the local excitation spectrum at an open boundary. Such information is of special interest in the case of topological phases where the boundary local density of states (LDOS) can reveal the presence of edge states or other type of localized excitations, thus allowing to check out the bulk-boundary correspondence and computing topological invariants [10, 11].

For this purpose we develop a method which allows to compute the bGF at a $d-1$ boundary starting from a $d$ dimensional infinite bulk. This method complements the approach of Refs. [12–14] for the calculation of bGFs by introducing the Faddeev-LeVerrier algorithm (FLA) [15–19], allowing to consider general $d$-dimensional systems with an arbitrary number of degrees of freedom and neighbours. This method performs the Fourier transform (FT)

needed to compute GFs with local components in the direction perpendicular to the junction to compute the bGF (see Fig. 1) by the analytic continuation of the momenta into the complex plane followed by residue integration. The FLA requires a low computational cost to obtain both the characteristic polynomial and the adjugate matrix of the secular equation $[\omega\hat{\mathbb{1}} - \hat{H}]$ in the same process which are, the main building blocks to solve the FT using the residue integration.

In addition to giving access to the spectral properties at the edges, bGFs allow the calculation of transport properties in heterostructures by means of non-equilibrium Green's function techniques [12, 20–24]. In addition the Green's function formalism allows to incorporate in a natural way electron-phonon and/or electron-electron interaction effects for example by means of diagrammatic techniques [25]. Even more, from bGFs it is possible to deduce effective Hamiltonians including all of these effects and obtain their topological properties [26–28].

In comparison with previous works, the semi-analytical approach used in Refs. [12–14] suffers from rigidity in the definition of the GF as it required analytical expressions for the key building blocks of the formalism, such as the characteristic polynomial. A typical symbolic Laplace expansion to evaluate the characteristic polynomial is highly inefficient with potentially enormous memory demand and computational complexity of $O(N!)$ [29] where $N$ is the total Hamiltonian dimension. This constraint imposes an upper-bound limit to the Hilbert space dimension that can be accounted by this method. In contrast, the only analytical entry required for the FLA is the polynomial decomposition of the Hamiltonian in the analytic continuation variable of the momentum perpendicular to the boundary $z = e^{ik_\perp L_\perp}$.

In the same spirit, the approach of Ref. [30] also uses the residue theorem to obtain the local components of the GF but taking a different path to get its poles. This method computes the poles by solving a generalized eigenvalue problem in contrast with our approach in which we obtain the poles as the roots of the characteristic polynomial. The main advantage of Ref. [30] approach is the possibility to work in the zero-limit for the broadening term, $\eta$, that enters as an imaginary part of the energy and the increased robustness of the code which circumvents the problem of obtaining the characteristic polynomial. In contrast, the residue integration of this approach must be modified by adding extra terms depending on the existence or not of boundary modes, and consequently, its mandatory to use a different routine to determine the existence or not of these localized modes.

Even though our approach is less robust and requires $\eta \neq 0$, it is quite transparent, flexible and easy to implement with a straightforward integration routine independent of the peculiarities of the system. Furthermore our approach exhibits better convergence performance compared to recursive approaches [31–34] for which precision is linked to the number of iterations [14].

The rest of the paper is organized as follows: in Sec. 2, we describe the computation of the Green's function formalism taking advantage of the residue theorem introduced in Refs. [14, 35]. We then use Dyson's equation to open a boundary in the bulk system with an infinity impurity perturbation. Sec. 3, we describe our method based on FLA to compute the boundary Green's functions with barely no analytical demands to operate. In Sec. 4, we use some relevant model Hamiltonians for 2D topological systems as examples to compute steadily the FLA, first in a purely analytic problem to then jump into purely computational approaches. These models include the 2D Chern insulator [36], the 2D Kitaev topological superconductor [37] and the Checkerboard lattice hosting topological flat bands [38]. Furthermore, we study the spectral properties at edges of such 2D models exhibiting topological features like chiral edge states. Sec. 5 includes a study of the convergence of the spectral density of the semi-infinite translational invariant Checkerboard lattice model comparing the recursive GF technique with the bGF obtained via FLA.

We finally summarize the main results with some conclusions in Sec. 6. Technical details

like the finite system diagonalization or an explicit FLA pseudocode are included in the appendices. Throughout, we use units with $nn$ hopping amplitude $t = 1$ and lattice parameter $a = 1$.

## 2 bGF method for 2D lattice models

To obtain the bGF we start from a $d$-dimensional bulk infinite system and introduce local perturbations with the characteristic profile that defines the boundary. As these local perturbations or impurity surface amplitude tends to infinity we are left with two $(d-1)$-dimensional open surfaces [7,8] e.g. two boundary lines in a 2D system induced by an impurity line, see Fig. 1. The bGF is obtained using the Dyson equations associated to the local surface impurity potential which breaks translational symmetry albeit the momenta in the direction parallel to the impurity surface are conserved and thus well defined.

The starting GF must be explicitly dependent on the local coordinate associated to the perpendicular direction to the boundary. In order to get these real space GFs, starting from $N \times N$ tight-binding Hamiltonians in momentum space $\hat{\mathcal{H}}(\mathbf{k})$, we have to compute the FT of the bulk GF in the direction perpendicular to the boundary. For this purpose, we decompose the momenta into parallel and perpendicular components $\mathbf{k} = (k_{\parallel}, k_{\perp})$ relative to the boundary direction (in higher dimensional models the parallel momentum component would be itself a vector $\mathbf{k}_{\parallel}$). The bulk Hamiltonian periodicity in both directions is set by $(L_{\parallel}, L_{\perp})$, such that $\hat{\mathcal{H}}(\mathbf{k} + 2\pi\mathbf{u}_{\perp}/L_{\perp}) = \hat{\mathcal{H}}(\mathbf{k})$, where $\mathbf{u}_{\perp}$ is the unitary vector in the perpendicular direction. As to compute the FT we need orthogonal lattice vectors, in some cases like the triangular lattice we have to double the primitive cell to obtain them. Using this periodicity, the Hamiltonian can be expanded in a Fourier series, $\hat{\mathcal{H}}(\mathbf{k}) = \sum_n \hat{\mathcal{V}}_n(k_{\parallel})e^{ink_{\perp}L_{\perp}}$, where $n$ covers the number of neighbours, and Hermiticity implies $\hat{\mathcal{V}}_{-n} = \hat{\mathcal{V}}_n^{\dagger}$. Then, the advanced bulk GF is defined as

$$\hat{G}^A(\mathbf{k}, \omega) = \left[ (\omega - i\eta)\hat{\mathbb{I}} - \hat{\mathcal{H}}(\mathbf{k}) \right]^{-1}, \tag{1}$$

where $\eta$ is a small broadening parameter that ensures the convergence of its analytic properties [39] (e.g. to compute the spectral densities and integrated quantities). This parameter is specially needed in the case of recursive methods where the spectrum is approximated by a finite set of poles. In this work we set $\eta = 2\Delta\omega/n_{\omega}$, where $\Delta\omega$ is the energy window that we are studying and $n_{\omega}$ is the number of points that we are computing within that window. The $N \times N$ matrix structure is indicated by the hat notation.

Fourier transforming along the perpendicular direction, the GF components are given by

$$\hat{G}^A_{jj'}(k_{\parallel}, \omega) = \frac{L_{\perp}}{2\pi} \int_{-\pi/L_{\perp}}^{\pi/L_{\perp}} dk_{\perp} e^{i(j-j')k_{\perp}L_{\perp}} \hat{G}^A(k_{\parallel}, k_{\perp}, \omega), \tag{2}$$

where $j$ and $j'$ are lattice site indices in the $x_{\perp}$−direction. By the identification $z = e^{ik_{\perp}L_{\perp}}$, this integral is converted into a complex contour integral,

$$\hat{G}^A_{jj'}(k_{\parallel}, \omega) = \frac{1}{2\pi i} \oint_{|z|=1} \frac{dz}{z} z^{j-j'} \hat{G}^A(k_{\parallel}, z, \omega). \tag{3}$$

Further simplification can be obtained by introducing the roots $z_n(k_{\parallel}, \omega)$ of the characteristic polynomial in the $z$−complex plane,

$$P(k_{\parallel}, z, \omega) = \det\left[ \omega\hat{\mathbb{I}} - \hat{\mathcal{H}}(k_{\parallel}, z) \right] = \frac{c_m}{z^m} \prod_{n=1}^{2m} \left( z - z_n(k_{\parallel}, \omega) \right), \tag{4}$$

where $m$ is the highest order of the characteristic polynomial and $c_m$ is the highest order coefficient. In terms of these roots the contour integral in Eq. (3) can be written as a sum over the residues of all roots inside the unit circle in the complex plane

$$\hat{G}^A_{jj'}(k_\parallel, \omega) = {\sum_{|z_n|<1}}' \frac{z_n^q \hat{M}(k_\parallel, z_n, \omega)}{c_m \prod_{l \neq n}(z_n - z_l)}, \tag{5}$$

where $q = j - j' + m - m' - 1$ and $z^{-m'}\hat{M}(k_\parallel, z, \omega)$ is the adjugate matrix of $[\omega\hat{\mathbb{1}} - \hat{\mathcal{H}}(k_\parallel, z)]$ where all the poles at zero were taken out of $\hat{M}$ as a common factor in $z^{-m'}$. Finally, $\sum'$ means that if $q < 0$ then we include $z_n = 0$ as a pole in the sum of residues (e.g., in the non local GF components with $j' > j$). Consequently when $q < -1$ higher order poles at zero appear in the sum of residues. To simplify these situations we can take advantage of the residue theorem to avoid these poles and compute the integral as

$$\hat{G}^A_{jj'}(k_\parallel, \omega) = -\sum_{|z_n|>1} \frac{z_n^q \hat{M}(k_\parallel, z_n, \omega)}{c_m \prod_{l \neq n}(z_n - z_l)}. \tag{6}$$

To simplify the notation, we omit the superscript 'A' denoting advanced GFs from now on. Given the real-space components of the bulk GF in Eq. (5), we next extend the method of Refs. [12,21,40] to derive the bGF characterizing a *semi-infinite* nearest-neighbour 2D systems. To this effect, we add an impurity potential line $\varepsilon$ localized at the frontier region. Taking the limit $\varepsilon \to \infty$ the infinite system is cut into two disconnected semi-infinite subsystems with $j \leq -1$ (left side, $L$) and $j \geq 1$ (right side, $R$), see Fig. 1. Using Dyson equation the local GF components of the cut subsystem follow as [12]

$$\hat{\mathcal{G}}_{jj} = \hat{G}^{(0)}_{jj} - \hat{G}^{(0)}_{j0}\left[\hat{G}^{(0)}_{00}\right]^{-1}\hat{G}^{(0)}_{0j}, \tag{7}$$

where $\hat{G}^{(0)}$ are the unperturbed bulk GF and $\hat{\mathcal{G}}$ are the semi-infinite perturbed GF. Following Eq. (7), the bGF for the left and right semi-infinite systems are respectively given by

$$\hat{\mathcal{G}}_L(k_\parallel, \omega) = \hat{\mathcal{G}}_{\bar{1}\bar{1}}(k_\parallel, \omega), \quad \hat{\mathcal{G}}_R(k_\parallel, \omega) = \hat{\mathcal{G}}_{11}(k_\parallel, \omega), \tag{8}$$

where the over-line in the local indices in the bGF means negative sites. In order to extend the method to the case of interactions to an arbitrary number of neighbours we need to include as many impurity surfaces as the number of neighbours in order to completely disconnect two semi-infinite regions. In an equivalent fashion we could obtain a new Dyson equation to compute the bGF of the system. Another possibility would be to define a supercell that transforms the Hamiltonian into a nearest-neighbour one with respect to this supercell and compute the bGF applying Eq. (8). Using this bGF we can compute the spectral properties of open (semi-infinite or finite) systems encoded in the spectral densities and the local density of states respectively

$$\rho_{L,R}(k_\parallel, \omega) = \frac{1}{\pi}\mathfrak{I}\,\mathrm{tr}\left\{\hat{\mathcal{G}}_{L,R}(k_\parallel, \omega)\right\}, \quad \langle\rho_{L,R}(\omega)\rangle = \int \frac{dk_\parallel}{\Omega_{k_\parallel}}\,\rho_{L,R}(k_\parallel, \omega), \tag{9}$$

where $\Omega_{k_\parallel} = 2\pi/L_\parallel$ accounts for the limits of integration.

## 3 Faddeev-LeVerrier algorithm

We first summarize FLA for a generic complex matrix. Let $\hat{A}$ be a $N \times N$ matrix with characteristic polynomial $P(\omega) = \det[\omega\hat{\mathbb{1}} - \hat{A}] = \sum_{k=0}^n \bar{C}_k \omega^k$. The trivial coefficients are $\bar{C}_n = 1$

## bGF COMPUTATION

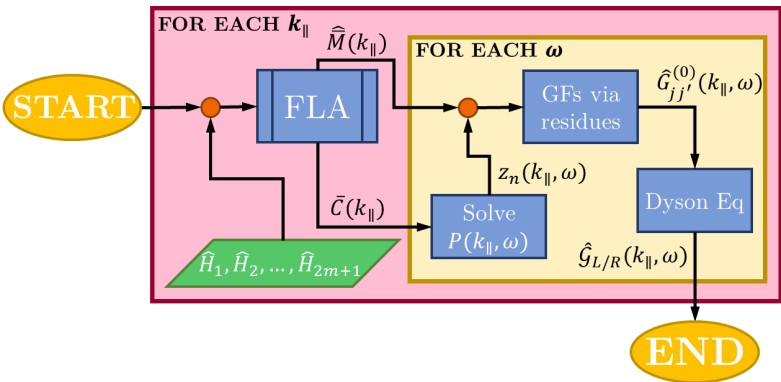

Figure 2: Complete algorithm workflow to compute the bGF using the FLA where the main input is the polynomial decomposition of the Hamiltonian for a given set of momenta $k_{\parallel}$ and frequencies $\omega$.

and $\bar{C}_0 = (-1)^n \det \hat{A}$, also simple is the term $\bar{C}_{n-1} = -\text{tr}\{\hat{A}\}$. The other coefficients can be calculated using the Faddeev-LeVerrier algorithm [15–19] as

$$\hat{\bar{M}}_k = \hat{A}\hat{\bar{M}}_{k-1} + \bar{C}_{n-k+1}\hat{\mathbb{1}}, \quad \bar{C}_{n-k} = -\frac{1}{k}\text{tr}\left\{\hat{A}\hat{\bar{M}}_k\right\}, \tag{10}$$

where $\hat{\bar{M}}_k$ is an auxiliary matrix such that $\hat{\bar{M}}_0 = 0$. *Remarkably* the matrices $\hat{\bar{M}}_k$ allow us to obtain the adjugate matrix of $[\omega\hat{\mathbb{1}} - \hat{A}]$ as a polynomial

$$\text{adj}\left[\omega\hat{\mathbb{1}} - \hat{A}\right] = \sum_{k=0}^{n} \omega^k \hat{\bar{M}}_{n-k}, \tag{11}$$

which, given that $\hat{\bar{M}}_0 = \hat{0}$, the adjugate matrix $\hat{M}(\omega)$ has $N-1$ order in $\omega$.

In our case $\hat{A} \equiv \hat{\mathcal{H}}(z)$ is a polynomial complex matrix and it can also be expanded as a polynomial in $z$ as

$$\hat{\mathcal{H}}(z) = \sum_{i=1}^{2m+1} \hat{H}_i z^{i-(m+1)} = \hat{H}_1 z^{-m} + \cdots + \hat{H}_{m+1} + \cdots + \hat{H}_{2m+1} z^m. \tag{12}$$

In some simple cases where $\text{rg}(\hat{H}_{2m+1}) = N$ we get the highest order polynomial decomposition for the Hamiltonian and $m = n_n N$ where $n_n$ is equal to the number of neighbours in the tight-binding model, but in general $m \leq n_n N$.

We are interested in expressing $\bar{C}_k$ and $\hat{\bar{M}}_k$ as two-variable polynomials in $\omega$ and $z$ using two variable FLA [41–43] to compute the complex integral. Still we have $\bar{C}_n = 1$ and $\hat{\bar{M}}_1 = \hat{\mathbb{1}}$. Then,

$$\bar{C}_{n-1}(z) = -\text{tr}\left\{\hat{\mathcal{H}}(z)\right\} = \sum_{i=1}^{2m+1} \bar{C}_{n-1,i} z^{i-(m+1)}. \tag{13}$$

For example, the next coefficients are

$$\hat{\bar{M}}_2(z) = \hat{\mathcal{H}}(z) + \bar{C}_{n-1}\hat{\mathbb{1}} = \sum_{i=1}^{2m+1} \hat{\bar{M}}_{2,i} z^{i-(m+1)},$$

$$\bar{C}_{n-2}(z) = -\frac{1}{2}\text{tr}\left\{\hat{\mathcal{H}}(z)\hat{\bar{M}}_2(z)\right\} = \sum_{i=1}^{4m+1} \bar{C}_{n-2,i} z^{i-(2m+1)}. \tag{14}$$

In this way we could get

$$\hat{\bar{M}}_k(z) = \sum_{i=1}^{2m(k-1)+1} \hat{\bar{M}}_{k,i} z^{i-(m(k-1)+1)}, \quad \bar{C}_{n-k}(z) = \sum_{i=i}^{2mk+1} \bar{C}_{n-k,i} z^{i-(mk+1)}, \qquad (15)$$

and deduce an explicit decomposition of $\text{adj}[\omega\hat{\mathbb{I}} - \hat{\mathcal{H}}(k_\parallel, z)]$ in $z$ from which we can extract the zero poles of the adjugate matrix $z^{-m'}$ as in Eq. (5). In simple cases where $m = n_n N$, it is straightforward to see that $m' = N - 1$.

In Fig. 2 we expose the general structure of the complete algorithm to compute the bGF given, as an input, the polynomial decomposition of the Hamiltonian particularized at any $k_\parallel$. Using FLA we obtain the auxiliary matrix to compute the adjugate of the secular equation $\hat{\bar{M}}(k_\parallel)$ and the coefficients of the characteristic polynomial $\bar{C}(k_\parallel)$, see Appendix B. From $\bar{C}(k_\parallel)$ we can compute the characteristic polynomial $P(k_\parallel, \omega)$ for any desired frequency and solve it to obtain the roots $z_n(k_\parallel, \omega)$.

Both $z_n(k_\parallel, \omega)$ and $\hat{\bar{M}}(k_\parallel)$ are the key ingredients to compute the unperturbed GFs in real space using Eq. (6) and taking as poles the roots that satisfy that $|z_n(k_\parallel, \omega)| > 1$. The order of the zero poles $m$ and $m'$ are totally determined by the polynomial decomposition in $z$ of $\bar{C}(k_\parallel)$ and $\hat{\bar{M}}(k_\parallel)$ respectively. Finally, we use Dyson equation to compute the bGFs of the system from the unperturbed ones.

## 4 Tight-binding models

In order to illustrate our method in a transparent self-explanatory way we take the example of common, well-known 2D topological Hamiltonians to compute the bGF explicitly. First, we start with the fully analytical $2 \times 2$ Chern insulator model [36] hosting chiral edge states to easily follow the FLA step by step. Later we consider more intricate examples where we have to partially or totally take advantage of the computational power of the FLA. These models include the 2D Kitaev model [37] for a topological superconductor showing Majorana edge modes and the 2D Checkerboard model which hosts topological flat bands with chiral edge states [38]. All these examples are relevant models for the study of topological matter in 2D and thus we exhibit the spectral density and the LDOS for an open boundary semi-infinite system to make explicit their topological edge properties. In Fig. 3 a) we show the Brillouin zone (BZ) for all of these different lattice models.

### 4.1 Chern insulator

We first illustrate the FLA with the well-known $2 \times 2$ Chern insulator Hamiltonian [36] in a square lattice described by

$$\hat{\mathcal{H}}(\mathbf{k}) = (M - \cos k_y - \cos k_x)\sigma_z + \sin k_x \sigma_x + \sin k_y \sigma_y, \qquad (16)$$

where $\sigma_\mu$ with $\mu = x, y, z$ are the Pauli matrices and $M$ is the mass term.

We FT along $k_x = k_\perp$ thereby we made the analytic continuation $z = e^{ik_x}$. We can now obtain the polynomial expansion of the Hamiltonian in $z$ following Eq. (12) where

$$\hat{H}_1 = \hat{H}_3^\dagger = (i\sigma_x - \sigma_z)/2, \quad \hat{H}_2 = (M - \cos k_y)\sigma_z + \sin k_y \sigma_y. \qquad (17)$$

We then compute the trivial $\bar{C}_n$ coefficients that define the characteristic polynomial in frequencies $(\omega)$

$$\bar{C}_2 = 1, \quad \bar{C}_1 = 0, \quad \bar{C}_0 = (M - \cos k_y)(z + z^{-1}) - [M^2 + 2(1 - M \cos k_y)], \qquad (18)$$

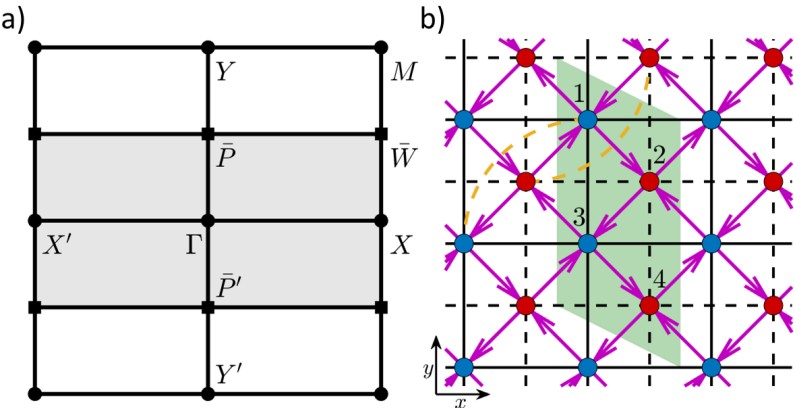

Figure 3: Brillouin zone for the square lattice models and real space representation of the Checkerboard lattice model. a) Square (white) and rectangular (grey shaded) BZ showing the high symmetry points in each one. The over-line in the high symmetry points denotes that they belong to the folded rectangular BZ in the $k_y-$direction. b) Checkerboard lattice. Red and blue dots indicate the sublattice sites. The magenta arrow, black dashed (solid) line and yellow dashed line accounts for the *nn* hopping $t$, the *nnn* hopping $t_1'$ ($t_2'$) and the *nnnn* hopping $t''$ respectively. The arrow direction shows the sign of the accumulated phase $\phi$ in the *nn* hopping terms. The shaded green region in b) corresponds to the doubling of the original primitive cell which produces the folding of the square BZ into a rectangular one as indicated in panel a).

consequently, their explicit decomposition in the $z$ polynomial

$$\bar{C}_{02} = \bar{C}_{04}^* = (M - \cos k_y), \quad \bar{C}_{03} = -[M^2 + 2(1 - M \cos k_y)]. \tag{19}$$

Due to the aforementioned relation, as $\mathrm{rg}(\hat{H}_{2m+1}) < N$, then $m < n_n N$ and for that we have a reduced degree of the characteristic polynomial obeying $\bar{C}_{01} = \bar{C}_{05} = 0$. Nevertheless, we have used the indexation of the polynomial in $z$ as the maximum degree polynomial for the sake of generalization of the method, similarly to the criteria taken in the pseudocode formulation in Appendix B.

Then, the characteristic polynomial takes the form

$$P(\omega) = (M - \cos k_y)(z + z^{-1}) + \omega^2 - M^2 - 2(1 - M \cos k_y), \tag{20}$$

where $c_m = \bar{C}_{04}$ and the non-trivial contributions to $\hat{\hat{M}}$ matrix are defined by

$$\hat{\hat{M}}_{21} = \hat{H}_1 + c_{11}\hat{\mathbb{1}} = \hat{H}_1, \quad \hat{\hat{M}}_{22} = \hat{H}_2 + c_{12}\hat{\mathbb{1}} = \hat{H}_2, \quad \hat{\hat{M}}_{23} = \hat{H}_3 + c_{13}\hat{\mathbb{1}} = \hat{H}_3. \tag{21}$$

Finally, the integral by residues for the bulk GF takes the form

$$\hat{G}_{jj'}(k_y, \omega) = -\frac{z_-^{j-j'}}{z_-} \frac{\begin{pmatrix} -(1+z_-^2) + \alpha z_- & i(1-z_-^2) - \beta z_- \\ i(1-z_-^2) + \beta z_- & (1+z_-^2) - \alpha z_- \end{pmatrix}}{2(M - \cos k_y)(z_- - z_+)}, \tag{22}$$

where $\alpha = 2(M + \omega - \cos k_y)$, $\beta = i2 \sin k_y$ and we have regularized the zeros of the adjugate matrix $\mathrm{adj}[\omega\hat{\mathbb{1}} - \hat{\mathcal{H}}(k_\parallel, z)]$ with the zeros of $P(\omega)$ knowing that $m = m' = 1$. Furthermore, we solve the trivial roots for $P(\omega)$ in Eq. (20), $z_\pm = (-b \pm \sqrt{b^2 - 4})/2$ where $b = [\omega^2 - M^2 - 2(1 - M \cos k_y)]/(M - \cos k_y)$ defining $|z_-| > 1$ and $|z_+| < 1$. Once the bulk GF in real space has been constructed we use Eq. (8) to obtain the corresponding bGFs.

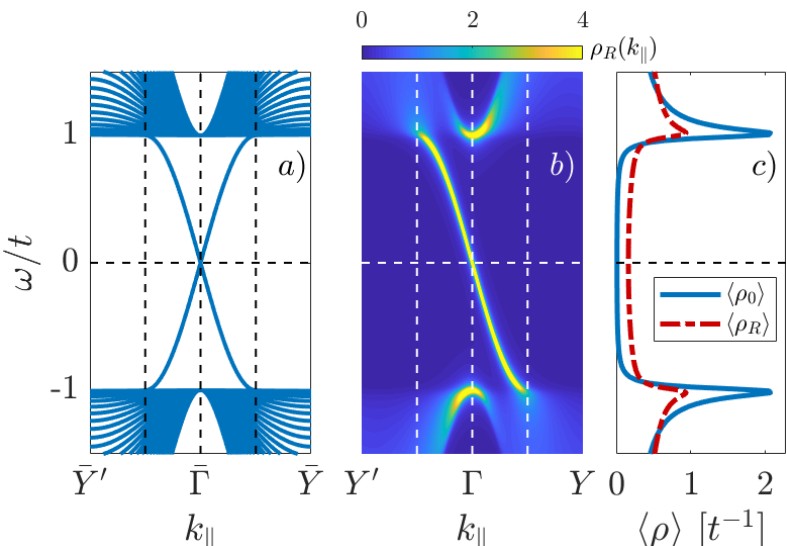

Figure 4: Open boundary characterization for the Chern Insulator model showing chiral edge states under the effect of the mass term $M \to 1$. a) Electronic bands obtained by exact diagonalization of a finite size system with $N_{sites} = 40$ sites. The spectrum shows 2 chiral edge states each one associated to a different boundary. b) Spectral density for a right boundary in the semi-infinite limit obtained from the bGF calculation. c) Integrated LDOS where straight (dot-dashed) line represents bulk (right boundary) LDOS.

In Fig. 4 we illustrate the open boundary spectral density for the topological phase of the Chern insulator exhibiting chiral edge states obtained using FLA. For comparison we also show the bands obtained using exact finite size Hamiltonian diagonalization, see Appendix A. As can be observed, while two chiral edge states are present in the finite system calculation, only one appears in the bGF calculation as expected for a semi-infinite system.

## 4.2 2D Kitaev square lattice

Now we apply FLA to obtain the characteristic polynomial of the $2 \times 2$ Kitaev square lattice model [37] and solve it computationally, in this way we can then obtain the bGF in a semi-analytic manner. The model Hamiltonian is given by

$$\hat{\mathcal{H}}(\mathbf{k}) = (\mu - \cos k_y - \cos k_x)\sigma_z - \Delta(\sin k_x + \sin k_y)\sigma_y, \tag{23}$$

where $\mu$ is the chemical potential and $\Delta$ is the pairing potential.

Again, the FT along $k_x = k_\perp$ is obtained using the analytic continuation $z = e^{ik_x}$. The polynomial expansion of the Hamiltonian in $z$ takes the expression

$$\hat{H}_1 = \hat{H}_3^\dagger = (-\sigma_z - i\Delta\sigma_y)/2, \quad \hat{H}_2 = (\mu - \cos k_y)\sigma_z + -\Delta \sin k_y \sigma_y. \tag{24}$$

We next compute the $\bar{C}_n$ coefficients that define the characteristic polynomial in powers of $\omega$ and $z$

$$\bar{C}_2 = 1, \quad \bar{C}_1 = 0, \quad \bar{C}_{01} = \bar{C}_{05}^* = (\Delta^2 - 1)/4, \quad \bar{C}_{02} = \bar{C}_{04}^* = (\mu - \cos k_y) - i\Delta^2 \sin k_y,$$

$$\bar{C}_{03} = \frac{(\Delta^2 - 1)}{2}\cos 2k_y + 2\mu \cos k_y - (1 + \Delta^2 + \mu^2), \tag{25}$$

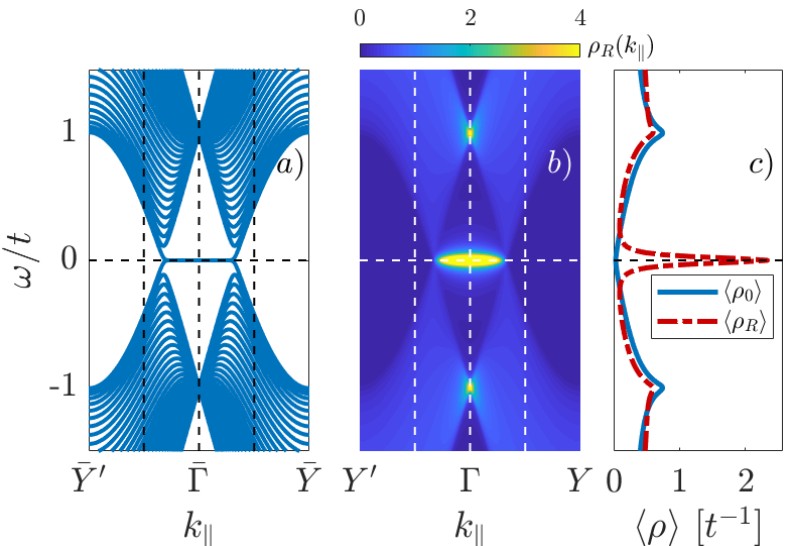

Figure 5: Open boundary characterization for the 2D Kitaev model showing Majorana flat band edge modes in the topological phase $\Delta = 1$ and $\mu = 1$. a) Electronic bands obtained by exact diagonalization of a finite size system with $N_{sites} = 40$ sites. The spectrum shows flat bands at both ends of the system. b) Spectral density for a right boundary in the semi-infinite limit obtained from the bGF calculation. c) Integrated LDOS where straight (dot-dashed) line represents bulk (right boundary) LDOS.

where $c_m = \bar{C}_{05}$ and finally the non-trivial contributions to the $\hat{\bar{M}}$ matrix are defined as

$$\hat{\bar{M}}_{21} = \hat{H}_1, \quad \hat{\bar{M}}_{22} = \hat{H}_2, \quad \hat{\bar{M}}_{23} = \hat{H}_3. \tag{26}$$

We regularize the zeros of the $\text{adj}[\omega\hat{\mathbb{I}} - \hat{\mathcal{H}}(k_\parallel, z)]$ with the zeros of $P(\omega)$ knowing that $m = 2$ and $m' = 1$. The integral by residues for the bulk GF takes the form

$$\hat{G}_{jj'}(k_y, \omega) = -2z_4^{j-j'} \frac{\begin{pmatrix} -(1+z_4^2) + \alpha z_4 & -\Delta[(1-z_4^2) - \beta z_4] \\ \Delta[(1-z_4^2) - \beta z_4] & (1+z_4^2) - \alpha z_4 \end{pmatrix}}{(\Delta^2 - 1)(z_4 - z_1)(z_4 - z_2)(z_4 - z_3)} + (z_4 \longleftrightarrow z_3), \tag{27}$$

where $\alpha = 2(\mu + \omega - \cos k_y)$, $\beta = i2\sin k_y$ and $|z_4|, |z_3| > 1$, thus $|z_2|, |z_1| < 1$. We omit the explicit analytical expression of the roots of the characteristic 4th degree polynomial due to their extension. As mentioned before, for this example it is convenient to obtain the roots computationally. In Fig. 5 we show typical results for the open boundary LDOS in the topological phase of the 2D Kitaev model showing Majorana flat band edge modes. Again, the comparison with the finite size diagonalization shows good agreement.

## 4.3 Flat band Checkerboard lattice

Finally we consider the $2 \times 2$ Checkerboard lattice model [38] which hosts topological flat bands and is defined by the Hamiltonian

$$\hat{\mathcal{H}}(\mathbf{k}) = \Omega_0(\mathbf{k})\hat{\mathbb{I}} + \Omega_1(\mathbf{k})\sigma_x + \Omega_2(\mathbf{k})\sigma_y + \Omega_3(\mathbf{k})\sigma_z, \tag{28}$$

where

$$\Omega_0(\mathbf{k}) = (t_1' + t_2')(\cos k_x + \cos k_y) + 4t'' \cos k_x \cos k_y, \quad \Omega_1(\mathbf{k}) = 4t \cos\phi \cos\frac{k_x}{2}\cos\frac{k_y}{2},$$

$$\Omega_2(\mathbf{k}) = 4t \sin\phi \sin\frac{k_x}{2}\sin\frac{k_y}{2}, \quad \Omega_3(\mathbf{k}) = (t_1' - t_2')(\cos k_x - \cos k_y). \tag{29}$$

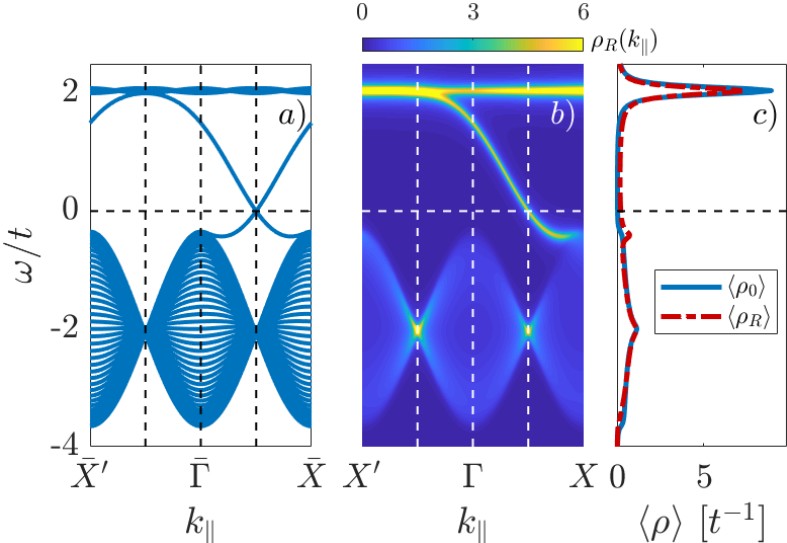

Figure 6: Open boundary characterization for the Checkerboard lattice model showing topological flat band at $\omega/t = 2$ with a chiral edge mode in the topological phase $\phi = -\pi/4$, $t'_1 = -t'_2 = t/(2+\sqrt{2})$ and $t'' = -t/(2+2\sqrt{2})$. a) Electronic bands obtained by exact diagonalization of a finite size system with $N_{sites} = 20$ sites with the unit cell doubled. The spectrum shows 2 chiral edge states each one associated to a different boundary. b) Spectral density for a right boundary in the semi-infinite limit obtained from the bGF calculation. c) Integrated LDOS where straight (dot-dashed) line represents bulk (right boundary) LDOS.

The system is thus characterized by *nn* hopping $t$, *nnn* hopping $t'_1$, $t'_2$ and *nnnn* hopping $t''$ terms, also the *nn* terms accumulate a phase $\phi$ pointed out in Fig. 3 b).

This model is an exemplification of a typical obstacle to tackle with our algorithm due to the sublattice degree of freedom. Due to that, the Hamiltonian includes lattice spacing fractions, hence if we try to FT with the analytic continuation $z = e^{ik_\perp L_\perp/2}$ instead of having a complex integral over the closed unit circle we arrive to an open arc integral in the complex plane, so we cannot apply the residue theorem to solve it. This kind of problems may also appear in Bravais lattices with non-orthogonal lattice vectors (e.g. the triangular lattice).

To circumvent this kind of obstacles we proceed to double the unit cell to obtain a new lattice with orthogonal lattice vectors and integer powers of $z = e^{ik_\perp L_\perp}$. The drawbacks of doubling the unit cell are that we are now working in a folded BZ and we have doubled the Hamiltonian degrees of freedom. Consequently the Hamiltonian in the new unit cell expressed in the basis $\Psi_{\mathbf{k}} = (\psi_{A1,\mathbf{k}}, \psi_{B2,\mathbf{k}}, \psi_{A3,\mathbf{k}}, \psi_{B4,\mathbf{k}})^T$ takes the form

$$\hat{\mathcal{H}}(\mathbf{k}) = \begin{pmatrix} \hat{A} & \hat{B} \\ \hat{B}^\dagger & \hat{A} \end{pmatrix}, \tag{30}$$

with

$$\hat{A} = \begin{pmatrix} \delta_2 & \beta_- \\ \beta_-^* & \delta_1 \end{pmatrix}, \quad \hat{B} = \begin{pmatrix} \alpha_1(1+e^{ik_y}) & e^{ik_y}\beta_+^* \\ \beta_+ & \alpha_2(1+e^{ik_y}) \end{pmatrix}, \tag{31}$$

where $\beta_\pm = e^{\pm i(k_x \pm \phi)} + e^{-i\phi}$, $\alpha_\mu = (t'_\mu + 2t''\cos k_x)$ and $\delta_\mu = 2t'_\mu\cos k_x$ with $\mu = 1, 2$.

In Fig. 3 b) we show the unit cell doubling in the $y$-direction for the Checkerboard lattice problem leading to a folded BZ along the $k_y$-direction. To avoid foldings in the spectral densities we have made the analytic continuation in $z = e^{ik_y}$ with $k_y = k_\perp$, in this way we have the explicit momenta dependence of the Hamiltonian in the unfolded BZ coordinate

$k_x = k_\parallel$. The polynomial expansion of the Hamiltonian in $z$ adopts the expression

$$\hat{H}_1 = \hat{H}_3^\dagger = \begin{pmatrix} 0 & 0 & 0 & 0 \\ 0 & 0 & 0 & 0 \\ \alpha_1 & 0 & 0 & 0 \\ \beta_+ & \alpha_2 & 0 & 0 \end{pmatrix}, \quad \hat{H}_2 = \begin{pmatrix} \delta_2 & \beta_- & \alpha_1 & 0 \\ \beta_-^* & \delta_1 & \beta_+ & \alpha_2 \\ \alpha_1 & \beta_+^* & \delta_2 & \beta_- \\ 0 & \alpha_2 & \beta_-^* & \delta_1 \end{pmatrix}. \tag{32}$$

Due to the cell doubling we have a characteristic off-diagonal representation of the $z$ dependent terms of the Hamiltonian which induces that $\mathrm{rg}(\hat{H}_{2m+1}) < N$, then again we have a degree reduction of the characteristic polynomial. We now could obtain analytically the $\bar{C}_n$ coefficients that define the characteristic polynomial but we omitted them due to their extension. These coefficients along with the adjugate matrix $\hat{M}(k_\parallel, z, \omega)$ can be obtained computationally in a straightforward way using Eq. (14), see Appendix B.

In Fig. 6 we show results for the open boundary LDOS for the topological phase of the Checkerboard lattice model exhibiting topological flat bands and chiral edge states. Again, the comparison with the bands obtained by direct diagonalization gives excellent agreement, except for the doubling of the edge states.

## 5 Comparison with recursive approaches

As mentioned before, the recursive GF method is a well established tool to compute bGFs. Below we briefly describe the recursive method taking advantage of the Hamiltonian decomposition into two perpendicular directions already introduced for FLA. We define the recursive method to compute the bGF at a dimensionless $n$-site as

$$\left[\hat{\mathcal{G}}_R^{rc}(n)\right]^{-1} = \omega\hat{\mathbb{1}} - \hat{\mathcal{H}}_0(k_\parallel) - \Sigma_R(n), \quad \left[\hat{\mathcal{G}}_L^{rc}(n)\right]^{-1} = \omega\hat{\mathbb{1}} - \hat{\mathcal{H}}_0(k_\parallel) - \Sigma_L(n), \tag{33}$$

where $\hat{\mathcal{H}}_0(k_\parallel)$ is the local contribution defined in each iteration step and the recursive expression of the self-energy takes the form

$$\Sigma_R(n) = \hat{T}_{LR}\left[\hat{\mathcal{G}}_R^{rc}(n-1)\right]^{-1}\hat{T}_{LR}^\dagger, \quad \Sigma_L(n) = \hat{T}_{LR}^\dagger\left[\hat{\mathcal{G}}_L^{rc}(n-1)\right]^{-1}\hat{T}_{LR}. \tag{34}$$

As can be observed, the self-energy at a given $n$-site couples this site with the previous one where $n$ goes from $n = 1$ to $n = N_{it}$ with $N_{it}$ is the number of recursive steps. The self-energy at the first site $\Sigma_{L/R}(n=1)$ can be defined to simulate the coupling to a doped continuum of the same material for better convergence.

From the polynomial decomposition of the Hamiltonian in Eq. (12) we can define the relevant matrices for the recursive method as

$$\hat{\mathcal{H}}_0 = \begin{pmatrix} \hat{H}_{m+1} & \hat{H}_m & \hat{H}_{m-1} & \cdots & \hat{H}_2 \\ \hat{H}_m^\dagger & \hat{H}_{m+1} & \hat{H}_m & \cdots & \hat{H}_3 \\ \hat{H}_{m-1}^\dagger & \hat{H}_m^\dagger & \hat{H}_{m+1} & \cdots & \hat{H}_4 \\ \vdots & \vdots & \vdots & \ddots & \vdots \\ \hat{H}_2^\dagger & \hat{H}_3^\dagger & \hat{H}_4^\dagger & \cdots & \hat{H}_{m+1} \end{pmatrix}, \quad \hat{T}_{LR} = \begin{pmatrix} \hat{H}_1^\dagger & \hat{H}_2^\dagger & \hat{H}_3^\dagger & \cdots & \hat{H}_m^\dagger \\ \hat{0} & \hat{H}_1^\dagger & \hat{H}_2^\dagger & \cdots & \hat{H}_{m-1}^\dagger \\ \hat{0} & \hat{0} & \hat{H}_1^\dagger & \cdots & \hat{H}_{m-2}^\dagger \\ \vdots & \vdots & \vdots & \ddots & \vdots \\ \hat{0} & \hat{0} & \hat{0} & \cdots & \hat{H}_1^\dagger \end{pmatrix}. \tag{35}$$

Notice that the dimension of the recursive method goes as $N_r = N n_n$ so for the usual $nn$ case satisfies $N_r = N$ and $\hat{\mathcal{H}}_0 = \hat{H}_2$, $\hat{T}_{LR} = \hat{H}_1^\dagger$.

In Fig. 7 we illustrate the convergence of the continuum spectrum within the recursive GF method for the Checkerboard model at $k_\parallel = \Gamma$ with parameters as in Fig. 6 for several number

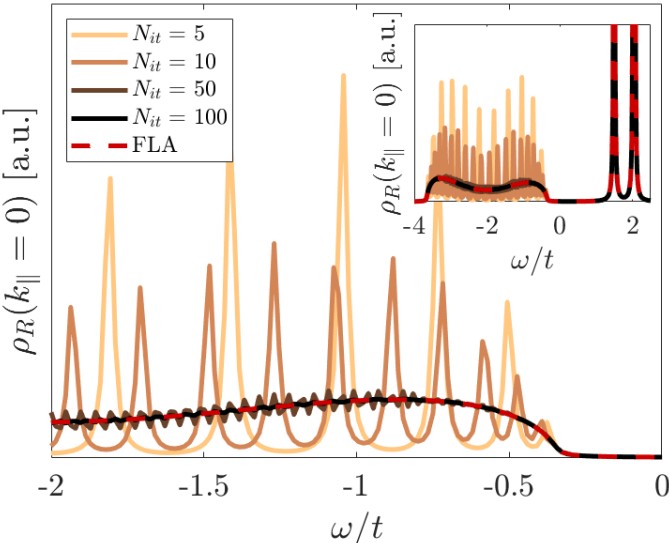

Figure 7: Open right boundary spectral density at $k_\parallel = \Gamma$ for the Checkerboard lattice model with $\eta = 0.02$ and the rest of parameters are the same as in Fig. 6. Solid lines represents the spectral density obtained by recursive GF for different number of recursive steps $N_{it} = 5, 10, 50$. Dashed red line is obtained using FLA. Main figure: top continuum valence bands contribution to the spectral density showing the discretization effect of the recursive method. Inset: All the contributions to the spectral density including the flat band at $\omega/t = 2$ and the topological chiral edge state at $\omega/t \approx 1.5$

of iterations compared to bGF obtained using FLA. While the recursive approach accounts well for discrete states, as boundary states, with few iterations, the number of recursive steps have to be greatly increased to properly converge the continuum spectrum into the semi-infinite limit [39]. In contrast, FLA provides an accurate description of both surface modes and continuum spectra without further computational effort. It is worth mentioning that the recursive method for all the lattice models in this publication takes from twice to four times more computing time than FLA for the same number of points in the spectral density and $N_{it} = 100$, for which, as shown in Fig. 7, the recursive calculation has not yet converged to a smooth continuum spectrum.

In order to compare the computational complexity of our technique one should have in mind that our method could be implemented in a partially analytical way, in the sense that we can provide an analytical expression for the characteristic polynomial for each of the cases that we study. The computational complexity is then limited to the evaluation of the roots of this polynomial which scales roughly as $O(M^2 \log M)$, where $M = 2m$ is the degree of the polynomial and the maximum degree possible is $M = 2m = 2n_n N$ (e.g., in a typical TB model up to $nn$, $M = 2N$ and for that its complexity goes as $\sim O(8N^2 \log N)$). On the contrary, the well-established recursive GF technique has $O(N_r^3 N_{it})$ complexity [39, 44], where $N_{it}$ typically $\gg 1$ is the number of iterations required for convergence in a desired energy precision $\eta$ and the term $N_r^3$ is due to matrix inversions where the recursive matrix dimension $N_r = N n_n$ grows with the number of neighbours.

For larger matrix dimensions or higher degree polynomials that the ones analyzed in this paper, FLA might suffer from numerical instability in the computation of the polynomial coefficients due to accumulated errors in the trace in Eq. (10) and from the recursive nature of the successive polynomial coefficients [45, 46]. However in Ref. [35] FLA was used to obtain the bGF of TB Hamiltonians that cannot be solved using symbolic approaches due to matrix

dimension (e.g., $N = 12$ Hilbert space dimension). So, despite the potential instability of the method, it still can be used to efficiently solve the bGF problem of TB Hamiltonians beyond analytical approaches, at least for moderate dimensions.

# 6 Conclusions and outlook

In this work we have extended the boundary Green function method developed in Refs. [14,35] to 2D lattices with hopping elements between arbitrary distant neighbors and solved the semi-analytical obstructions to compute the bGF for large systems, non-orthogonal lattice vectors or Hamiltonians with terms with momentum fractions. This was made by implementing the Faddeev-LeVerrier algorithm to compute the characteristic polynomial and the adjugate matrix, the building blocks to compute the bGF. As an illustration of the method we have analyzed the spectral properties of different topological 2D Hamiltonians showing the appearance of topological states.

With FLA we can compute the bGF for any TB model with a well-known algorithm and a simple implementation which provides the coefficients of the characteristic polynomial but also the adjugate matrix in the same process. Furthermore, FLA can be extended to obtain the generalized inverses of multiple-variable polynomials or particularly, two-variable polynomials [41–43].

In Ref. [47, 48] it is claimed that the classical Faddeev-LeVerrier algorithm for polynomial matrices in one variable has $O(N^3 N)$ computational complexity and it avoids any division by a matrix entry, which it is desirable from the convergence perspective in contrast to recursive approaches. Although the classical FLA is not the most efficient algorithm from the point of view of complexity (e.g. Berkowitz algorithm [49] is faster), it is a rather simple and general way to solve the inverse of a polynomial matrix problem. Despite the recursive nature of FLA, it can be easily modified to carry out the $N$ matrix multiplications in parallel [46,47,50,51].

As an outlook, the FLA method can be combined with interpolation approaches [48,52,53] to improve the stability of the algorithm when computing the bGF of TB systems with a large number of degrees of freedom and neighbours. Furthermore, this method and the recursive methods could be combined to describe systems with regions with broken translational symmetry like two semi-infinite translational invariant regions coupled by a disordered region. In addition, we foresee the application of the method to study higher order topological insulators [54] which requires projection onto the intersection of two or more edge surfaces.

# Acknowledgments

We acknowledge and thank P. Burset for useful comments on this manuscript. This project has been funded by the Spanish AEI through Grant No. PID2020-117671GB-I00 and through the "María de Maeztu" Programme for Units of Excellence in R&D (CEX2018-000805-M).

# A    Exact Hamiltonian diagonalization

From the matrices that define the recursive method in Eq. (35) we can also describe the total Hamiltonian for a finite system to compute an exact diagonalization and obtain the edge state

spectrum.

$$\hat{H}_{TOT} = \begin{pmatrix} \hat{\mathcal{H}}_0 & \hat{T}_{LR} & \hat{0} & \cdots & \hat{0} \\ \hat{T}_{LR}^{\dagger} & \hat{\mathcal{H}}_0 & \hat{T}_{LR} & \cdots & \hat{0} \\ \hat{0} & \hat{T}_{LR}^{\dagger} & \hat{\mathcal{H}}_0 & \cdots & \hat{0} \\ \vdots & \vdots & \vdots & \ddots & \vdots \\ \hat{0} & \hat{0} & \hat{0} & \cdots & \hat{\mathcal{H}}_0 \end{pmatrix}, \tag{36}$$

where the main diagonal has $N_{sites}$ block elements and total dimension $N_d = N_{sites} N n_n$ so for the usual $nn$ case satisfies $N_d = N_{sites} N$ and $\hat{\mathcal{H}}_0 = \hat{H}_2$, $\hat{T}_{LR} = \hat{H}_1^{\dagger}$.

## B  Faddeev-LeVerrier algorithm

We include here a simple pseudocode description of the classic FLA [15–19] to obtain the coefficients of the characteristic polynomial $\bar{C}$ and the polynomial description of the adjugate matrix $\hat{\bar{M}}$ of the secular equation $[\omega \hat{\mathbb{1}} - \hat{H}]$ from a constant matrix (Algorithm 1).

---

**Algorithm 1** Classic Faddeev-LeVerrier algorithm

---

**Input:** $\hat{H} \in \mathbb{C}^{n \times n}$ where $n \geq 2$
**Output:** $(\bar{C}, \hat{\bar{M}})$
 1:  $\bar{C}_n = 1$, $\hat{\bar{M}}_1 = \hat{\mathbb{1}}$, $k \leftarrow 2$
 2:  $\bar{C}_{n-1} = -\text{tr}\{\hat{H}\}$
 3:  **while** $k \leq n$ **do**
 4:      $\hat{\bar{M}}_k \leftarrow \hat{H}\hat{\bar{M}}_{k-1} + \bar{C}_{n-k+1}\hat{\mathbb{1}}$
 5:      $\bar{C}_{n-k} \leftarrow -\frac{1}{k}\text{tr}\{\hat{H}\hat{\bar{M}}_k\}$
 6:      $k \leftarrow k + 1$
 7:  **end while**

---

We also describe the modified FLA for two variable polynomials in $(\omega, z)$ where the matrix itself $\hat{\mathcal{H}}(z)$ is a polynomial matrix [41–43] given as an entry the polynomial decomposition in $z$ of the Hamiltonian as in Eq. (12) (Algorithm 2).

---

**Algorithm 2** Two-variable Faddeev-LeVerrier algorithm

---

**Input:** $\hat{H}_1, \hat{H}_2, \ldots, \hat{H}_{2m+1} \in \mathbb{C}^{n \times n}$ where $n \geq 2$
**Output:** $(\bar{C}, \hat{\bar{M}})$
1: $\bar{C}_n = 1$, $\hat{\bar{M}}_1 = \hat{\mathbb{1}}$, $k \leftarrow 2$
2: $\bar{C}_{n-1,1} = -\text{tr}\{\hat{H}_1\}$, $\bar{C}_{n-1,2} = -\text{tr}\{\hat{H}_2\}$, $\ldots$,
   $\bar{C}_{n-1,2m+1} = -\text{tr}\{\hat{H}_{2m+1}\}$
3: **while** $k \leq n$ **do**
4:     **for** $i \leftarrow 1 : 2m(k-1)+1$ **do**
5:         **if** $i \leq 2m(k-2)+1$ **then**
6:             $\hat{\bar{M}}_{k,i} \leftarrow \hat{\bar{M}}_{k,i} + \hat{H}_1 \hat{\bar{M}}_{k-1,i}$
7:         **end if**
8:         **if** $i \geq 2$ and $i \leq 2m(k-2)+2$ **then**
9:             $\hat{\bar{M}}_{k,i} \leftarrow \hat{\bar{M}}_{k,i} + \hat{H}_2 \hat{\bar{M}}_{k-1,i}$
10:         **end if**
11:         $\ldots$
12:         **if** $i \geq 2m+1$ and $i \leq 2m(k-2)+2m+1$ **then**
13:             $\hat{\bar{M}}_{k,i} \leftarrow \hat{\bar{M}}_{k,i} + \hat{H}_{2m+1} \hat{\bar{M}}_{k-1,i}$
14:         **end if**
15:         $\hat{\bar{M}}_{k,i} \leftarrow \hat{\bar{M}}_{k,i} + \bar{C}_{n-k+1}\hat{\mathbb{1}}$
16:     **end for**
17:     **for** $i \leftarrow 1 : 2mk+1$ **do**
18:         **if** $i \leq 2m(k-1)+1$ **then**
19:             $\bar{C}_{n-k,i} \leftarrow \bar{C}_{n-k,i} - \frac{1}{k}\text{tr}\{\hat{H}_1 \hat{\bar{M}}_{k,1}\}$
20:         **end if**
21:         **if** $i \geq 2$ and $i \leq 2m(k-1)+2$ **then**
22:             $\bar{C}_{n-k,i} \leftarrow \bar{C}_{n-k,i} - \frac{1}{k}\text{tr}\{\hat{H}_2 \hat{\bar{M}}_{k,i}\}$
23:         **end if**
24:         $\ldots$
25:         **if** $i \geq 2m+1$ and $i \leq 2m(k-1)+2m+1$ **then**
26:             $\bar{C}_{n-k,i} \leftarrow \bar{C}_{n-k,i} - \frac{1}{k}\text{tr}\{\hat{H}_{2m+1} \hat{\bar{M}}_{k,i}\}$
27:         **end if**
28:     **end for**
29:     $k \leftarrow k+1$
30: **end while**

---

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
