# Peer review of "D topological matter from a boundary Green's functions perspective: Faddeev-LeVerrier algorithm implementation"

_SciPost Physics, doi:SciPost Phys. 13, 009 (2022)_

## Round 1 · Referee Report · Anonymous (Referee 1) · 2021-10-20

Report

The manuscript under consideration introduces an improved variant of the so-called boundary Green’s function (bGF) method, which is presented as specifically suited for (numerically?) obtaining transport properties in heterostructures.

I am familiar with several of the established approaches that are mentioned (e.g. recursive Green's function method, wave matching). However, I find myself unable to assess this manuscript due to the problems listed under below under "requested changes".

In addition to the requested changes, I would also like to rise the following point: Is an example implementation of the new method available to interested readers? Ideally, in the spirit of open access, a working example implementation would accompany the manuscript.

Once the "requested changes" have been addressed I will be hopefully able to properly review this manuscript.

Requested changes

  1. The manuscript needs to define "bGF". Section 2 begins with the words "To obtain the bGF we start" but it is never defined. This is, after all, the central concept of the text.

  2. More generally, I think that it should be clearer what this article is about to a person who has research experience in the field but might be not already familiar with the existing bGF literature by the authors. What kind of systems does the method apply to? What kind of properties can be computed? Is the new method suitable for computing mundane transport properties like the conductance of a disordered heterostructure?

  3. The authors claim that their new method of computing transport properties is in some ways superior to established methods. This should be demonstrated clearly and quantitatively. Section 5 contains a comparison to "standard recursive Green’s function calculations" but I struggle to recognize therein what is commonly known as the recursive Green's function approach [1] in the quantum transport community.

[1] A. MacKinnon, Zeit. f. Phys. B 59, 385 (1985).

  • validity: -
  • significance: -
  • originality: -
  • clarity: low
  • formatting: perfect
  • grammar: good

Author:  Miguel Alvarado  on 2021-10-22  [id 1872]

(in reply to Report 1 on 2021-10-20)

We thank the Referee for his/her disposition to review our manuscript. We provide below the requested information which would hopefully allow him/her to complete a report.

In the first place, we agree that an example implementation available to interested readers would be desirable. We have thus uploaded to the Zenodo repository the Matlab codes needed to accomplish the Faddeev-Leverrier algorithm for the Chern Insulator as in Fig. 4 in the manuscript (see https://doi.org/10.5281/zenodo.5593119). This link will be provided in the manuscript when revision be allowed by the editor, after this first round of referees.

Regarding the requested further information on the boundary Green’s function method, we plan to revise our manuscript in order to give further explanations on it and on its relationship with other methods to obtain transport properties based on Green’s function techniques. Let us try to summarize here what we believe could be the origin of some possible misunderstanding.

Boundary Green’s functions (the definition of bGF appears at the third paragraph of the introductory section) encode information on the local spectral properties of semi-infinite systems. Such information is of special interest, for instance, in the case of topological phases where the boundary spectral densities can reveal the presence of edge states or other type of localized excitations. Additionally, bGFs are the basic input to compute general transport properties of different type of junctions within the so-called “boundary Green’s function method”, which is based on non-equilibrium Green’s function techniques as explained in Refs. [10-15] of the manuscript. From the Referee comments we realize that the idea on this method might not be clearly stated within the manuscript. In particular, we have realized that we should emphasize that this method is not appropriate to calculate transport properties of extended disordered system as the one described in the reference pointed out by the Referee, but is rather better suited for the case of short junctions.

Extended disordered systems could be, of course, studied using recursive Green’s function techniques which do not require any kind of translational invariance. On the other hand, recursive Green’s function techniques can be applied to obtain the bGF of semi-infinite translational invariant systems and that is what the comparison by the end of the manuscript is about.

Finally let us comment that our FLA method and the recursive methods could be combined to describe systems where two semi-infinite translational invariant regions are coupled by a disordered region.

We plan to revise our manuscript in order to clarify the above mentioned issues and also add a reference to the manuscript pointed out by the referee in connection to recursive Green’s functions calculations. We are now fully open to consider any other possible suggestion by the Referee.

---

## Round 1 · Referee Report · Adrian Feiguin (Referee 2) · 2021-11-22

Strengths

1- This work builds upon previous papers by the authors and collaborators where the boundary Green's function is introduced in 1D, by extending the formalism to arbitrary dimensions and using the Faddeev-LeVerrier algorithm to solve for the Green's function.

2- The authors claim to show a considerable improvement compared to recursive methods in terms of stability and computational cost.

3- They demonstrate the method with several applications to 2D (quadratic) topological Hamiltonians.

4- The method is described in great details, including pseudo-code.

5- The formalism is beautiful and elegant. It can become a standard tool in the study of heterostructures within ab-initio frameworks. I recommend the authors cite relevant references in this context such as: - Haydock R. The recursive solution of the Schrödinger equation. Comput Phys Commun. (1980) 20:11–6. -Viswanath VS, Müller G. The Recursion Method: Application to Many Body Dynamics. Berlin; Heidelberg: Springer (1994).

Weaknesses

1- The authors briefly mention in the introduction that the method can be extended to problems with electron-electron interactions. The paper is self-contained and I would not request an example of such a calculation, but it would be useful to understand how the formalism would be extended to a case in which one has quartic terms. Maybe a descriptive paragraph in the conclusions would suffice.

Report

I recommend it for publication in SciPost after my aforementioned comments are addressed.
  • validity: top
  • significance: high
  • originality: high
  • clarity: high
  • formatting: perfect
  • grammar: excellent

Author:  Miguel Alvarado  on 2021-11-26  [id 1973]

(in reply to Report 2 by Adrian Feiguin on 2021-11-22)

We thank the referee for his/her disposition to review our manuscript. We provide below the requested information which would hopefully allow him/her to complete a report.

In the main text we quote some references regarding the use of GF to compute magnetic interaction or to deduce effective Hamiltonians for such systems [1-3]. On the other hand, Green’s functions provide the appropriate starting point to include interactions by means of diagrammatic techniques. We shall include a comment along these lines in the revised manuscript.

[1] Z. Wang and S.-C. Zhang, Simplified topological invariants for interacting insulators, Phys Rev. X2, 031008 (2012), doi:10.1103/PhysRevX.2.031008
[2] M. Iraola, N. Heinsdorf, A. Tiwari, D. Lessnich, T. Mertz, F. Ferrari, M. H. Fischer, S. M. Winter, F. Pollmann, T. Neupert, R. Valentí and M. G. Vergniory, Towards a topological quantum chemistry description of correlated systems: the case of the hubbard diamond chain, arXiv preprint arXiv:2101.04135 (2021),2101.04135.
[3] D. Lessnich, S. M. Winter, M. Iraola, M. G. Vergniory and R. Valentí, Elementary band representations for the single-particle green’s function of interacting topological insulators, Physical Review B104(8) (2021), doi:10.1103/physrevb.104.085116

---

## Round 2 · Referee Report · Anonymous (Referee 1) · 2022-2-9

Report

Warnings issued while processing user-supplied markup:

  • Inconsistency: Markdown and reStructuredText syntaxes are mixed. Markdown will be used.
    Add "#coerce:reST" or "#coerce:plain" as the first line of your text to force reStructuredText or no markup.
    You may also contact the helpdesk if the formatting is incorrect and you are unable to edit your text.

In my first report I stated:

  1. The manuscript needs to define "bGF". Section 2 begins with the words "To obtain the bGF we start" but it is never defined. This is, after all, the central concept of the text.

In their response to this report, the authors claim that a definition was already present in the manuscript: “the definition of bGF appears at the third paragraph of the introductory section”. They refer to the following paragraph of version 1:

In this work we focus on the boundary Green’s function (bGF) method, which is specifically suited to obtain transport properties in heterostructures [10–15]. The bGF approach allows also to explore electronic spectral properties such as the local density of states (LDOS) or checking out the bulk-boundary correspondence of topological phases and computing topological invariants [16,17]. Furthermore, the Green’s function formalism allows to incorporate in a natural way electron-phonon and/or electron-electron interaction effects. Even more, from bGFs it is possible to deduce effective Hamiltonians including all of these effects and obtain their topological properties [18–20].

I read this paragraph carefully multiple times before writing my first report. This is not a definition, not even a hand-waving one. There is not even a precise pointer to where such a definition could be found. Instead, the authors cite a series of six articles by themselves (Refs. 10 to 15). Are interested readers to look up all these six articles to even see what the current one is about?

I maintain that an article on “boundary Green’s functions” should contain an appropriate introduction of this concept. In an substantial article of 20 pages, half a page of text including a few equations could be certainly devoted to an introduction of the central concept. Such an introduction should have an appropriate level of mathematical rigor and contain specific pointers to more in-depth information.

In version 2 of the manuscript the paragraph has been expanded:

In this work we focus on the boundary Green’s function (bGF) method, which is specifically suited to obtain transport properties in heterostructures based on non-equilibrium Green’s function techniques [10–15]. The bGFs encode information on the local spectral properties of semi-infinite regions. In addition to transport, such information is of special interest, for instance, in the case of topological phases where the boundary local density of states (LDOS)s can reveal the presence of edge states or other type of localized excitations, thus it is possible to check out the bulk-boundary correspondence of topological phases and computing topological invariants [16, 17]. Furthermore, Green’s function formalism allows to incorporate in a natural way electron-phonon and/or electron-electron interaction effects for example by means of diagrammatic techniques [18]. Even more, from bGFs it is possible to deduce effective Hamiltonians including all of these effects and obtain their topological properties [19–21].

However, in my opinion, this still does not define what “bGFs” are, it merely lists some properties and applications.

Looking at the examples and the provided references I have the impression that the “bGF” is about computing the Green’s function of a 2d bulk that is terminated by a 1d boundary, in other words a half-bulk. But is this true? A clear problem statement and perhaps even a helpful figure would go a long way to show immediately what this work is about.

I find that my question number 3 has not yet been addressed. As I tried to point out in my first report, at least within the quantum transport community, the term “recursive Green’s function method” is used for the technique pioneered by A. MacKinnon in Zeit. f. Phys. B 59, 385 (1985). The purpose of this technique is the computation of the retarded Green’s function within a finite scattering region to which semi-infinite quasi-1d leads have been attached.

However when the authors refer to “standard recursive Green’s function calculations” (see abstract) they seem to mean a different recursive Green's function method that is a “well established tool to compute bGFs” (see section 5). Clarifying the terminology unambiguously would make the article easier to understand for researchers with a background in quantum transport.

It’s laudable that the authors agreed to share the source code associated with the article. Unfortunately, the provided material consists of four matlab files without any further explanation. A short readme file that gives an overview of the different files and functions would be very useful if the interested reader is not supposed to reverse-engineer the code.

I tried to run the code with Octave (a free clone of matlab), but while the computation seems to finish (the progress indicator reaches 100%), the script fails subsequently with an error. ( Clearly, since the program is meant to be run with Matlab, this is not a fault, but the authors might be interested about this.)

Requested changes

  1. Properly define “bGF”, the central concept of this article.

  2. Clarify the relation to “standard recursive Green’s function calculations”.

  3. Provide a problem statement that is clear to people who are new to “bGF” formalism: what is the purpose and current status of the “bGF” formalism? How does it compare to other approaches, for example the one introduced in Phys. Rev. Research 1, 033188 (2019)? What is the improvement brought about by the current article? (The introduction mentions “diverse disadvantages of the semi-analytical calculations” but this is very vague.)

In my opinion addressing these issues requires substantially reworking the introductory section.

  • validity: good
  • significance: -
  • originality: -
  • clarity: low
  • formatting: perfect
  • grammar: good

Author:  Miguel Alvarado  on 2022-02-21  [id 2231]

(in reply to Report 1 on 2022-02-09)

We thank the referee for his/her disposition to review our manuscript. We provide below the requested information.

  1. In our last revision of the manuscript, we have tried to give further information on the bGF method. Now we understand that the Referee is asking to provide a direct definition of the bGF concept. As the mathematical definition is given in Section 2, what we have done is to add a qualitative definition in the introduction as the quantity which “encodes the local excitation spectrum at an open boundary” of semi-infinite system. We have furthermore restructured the introduction following the recommendation to focus on the purpose and status of the method, and pointing out the advantages with respect to other approaches.

  2. The bGF can be computed using recursive methods, as described in Section 5 of the manuscript. As there exist a large variety of such methods which have been used for many decades within the quantum transport community we agree that there could be some ambiguity regarding what we mean by “standard recursive techniques”. We have thus removed the term “standard” in the abstract and refer the reader to the explicit definition of the recursive method, which is given in Section 5.

  3. As mentioned in 1., we have rewritten the introduction to made it more clear not only the bGF concept but also what we mean by the “bGF method” for transport calculations. We hope that in this new writing we made clear the distinction between the method that we develop for the bGF calculation and the “bGF method” for transport calculations, which is something independent on how one obtains the bGF. We also explicitly mention the relation of our method to that described in Phys. Rev. Research 1, 033188 (2019). This work also uses the residue theorem to obtain the Fourier transform of the GF but takes a different path to obtain its poles GF in momentum space, by solving a generalized eigenvalue problem. Another important difference between the approaches is that in our method a non-zero broadening term (eta) that enters as an imaginary part of the energy is needed in order to simplify the integration by residues. In the PRR (2019) they must modify the residue integration by adding extra terms depending on the existence or not of boundary modes, and consequently, it’s mandatory to use a different routine to determine the existence or not of these localized modes. Even though our approach is less robust and works in the limit eta ≠ 0, it is quite transparent and easy to implement with a straightforward integration routine independent of the particularities of the Hamiltonian. We also explain the advantage of our method with respect previous semi-analytical approaches that demand the analytical expression of the coefficients of the characteristic polynomial. These other approaches are limited to small Hilbert spaces since a typical symbolic Laplace expansion to evaluate the characteristic polynomial is highly inefficient with computational complexity of O(N!).

  4. We have included a README file in the code repository. In relation to the execution of our code in Octave we have found that the problem mentioned by the Referee is related to the visualization part of the subroutine. We have rewritten the plotting subroutine to facilitate the execution of the code by open-source users.

---

## Round 2 · List of Changes

Dear editor,

Thank you for the referee reports on our manuscript. We have answered them separately and modified the manuscript in accordance to their comments. The list of changes in the manuscript is provided below.

  1. We have uploaded to the Zenodo repository the Matlab codes needed to accomplish the Faddeev-Leverrier algorithm for the Chern Insulator as in Fig. 4 in the manuscript.

  2. We include in the introductory section further explanations on the kind of systems that can be studied from the boundary Green’s function perspective and the general properties that can be computed.

  3. Also in the introduction we mention the possibility to treat interactions diagrammatically using Green’s functions, giving a reference to a particular example treating the electron-phonon interaction.

  4. We include an explanatory paragraph in section II to properly define the boundary Green’s function for systems with an arbitrary number of neighbours.

  5. In the conclusions we add a comment to clarify that our method can be combined with recursive approaches to accomplish finite regions without translational symmetry (e.g. disordered heterostructures).

  6. We have included in the manuscript the references suggested by both referees in connection to recursive Green’s function techniques.

---

## Round 3 · Referee Report · Anonymous · 2022-6-3

Report

This article is part of a series of works by the authors in which they develop a numerical method called boundary Green's function method. In this particular paper the method is refined by the use of the Faddeev-LeVerrier algorithm.

The article demonstrates the application of the method to compute the spectral density of a 2d-half-bulk described by a tight-binding Hamiltonian.

I think that this method could be valuable to interested researchers in the field and therefore recommend publication in SciPost.

After two rounds of review I find the article to be now sufficiently clear, although in my opinion there remains room for improvement. In particular, I find the initial introduction and description of the method still too vague. It is only when looking at applications in section 4 that one develops a concrete understanding of the method.

---

## Round 3 · List of Changes

Dear editor,
Thank you for the referee reports on our manuscript.
1. We have modified the introduction of the manuscript in accordance with their comments to clarify the main objective of the paper, the conceptual basis of it and the purpose of the method compared to other approaches.

---

## Editorial Decision

published